# Four-Qubit Cluster States Generation through Multi-Coin Quantum Walk

**Tianyi Wang, Xiaoguang Chen *** and **Jianxiong Liang**

Department of Communications Science and Engineering, School of Information Science and Technology, Fudan University, Shanghai 200433, China
\* Correspondence: xiaoguangchen@fudan.edu.cn or xgchen@fudan.ac.cn

**Abstract:** Quantum computing requires large numbers of resources of entangled qubits, which cannot be satisfied using traditional methods of entanglement generation, such as optical systems. Therefore, we need more efficient ways of entanglement generation. It has been proved that multi-coin quantum walks can be used to replace direct Bell state measurements during the process of entanglement generation in order to avoid the difficulty of Bell state measurement. In this paper, we take one step further and generate 4-qubit cluster states using multi-coin quantum walks, which simplifies the generation of 4-qubit cluster states by using only Bell states and local measurements. We also propose a method for preparing 4-qubit cluster states with quantum circuits to facilitate their use in quantum computing.

**Keywords:** cluster states; multi-coin quantum walk; quantum gate; quantum circuit

## 1. Introduction

With the development of quantum information technology, quantum computing is becoming more demanding and realistic than ever. Traditional quantum computing is based on quantum circuits composed of single-qubit and multi-qubit quantum gates, which is difficult to implement in real world. As an improvement of traditional quantum computing, Briegal and Raussendorf et al. proposed the concept of measurement-based quantum computing (MBQC) in 2001 [1,2]. Although MBQC simplifies the process of quantum computing, it requires an efficient way to generate cluster states to carry out the operations.

Cluster states, introduced by Nielsen in 2004 [3], are multiple qubits that are highly entangled. Compared with other types of entangled states, cluster states are easier to maintain their entangled properties, which is very suitable for quantum computing. Cluster states are traditionally generated based on optical systems and requires a large number of repetitive operations. Improved methods have been proposed: Browne et al. proposed the Browne–Rudolph protocol in 2005 [4], which can generate new cluster states with a probability of 0.5 by using two types of fusion gates while consuming one photon; Gilbert et al. proposed an improved Browne–Rudolph protocol in 2006 [5]; Louis et al. proposed a new method for generating cluster states using weak non-linearity in 2007 [6]. However, these methods are still too complicated to be implemented in real applications. Therefore, easier and more efficient methods of cluster state generation are required.

Multi-coin quantum walk can be used to address this problem. Coined quantum walk is one kind of discrete-time quantum walk that combines the freedom of qubits with the classical random walk and defines a new type of quantum coin [7,8]. Additionally, multi-coin quantum walk involves several coin operators that manipulate the qubits, which can be implemented to carry out specific operations [9–12]. Recent researches have proved that quantum walk-like protocols can be used to carry out entanglement swapping [13] and multi-coin quantum walk can generate 2-qubit entangled state and 3-qubit GHZ state using only Bell states and local measurement [14]. It has also been proved that coined

quantum walk can be used to construct quantum gates and implement universal computing in 2-qubit ot 3-qubit systems [15]. In this paper, we extend the method proposed in [14] and generate 4-qubit cluster states between targeted qubits with 3 pairs of Bell states and local measurement, which significantly simplifies the process of traditional cluster state generation. We also propose a practical implementation of 4-qubit system of quantum gates and construct a quantum circuit using multi-coin quantum walk to generate 4-qubit cluster state. This method provides a more physically feasible option for real applications.

This paper is organized as follows. First, we briefly introduce the process of multi-coin quantum walk and describe the generation of 3-qubit GHZ state and 4-qubit cluster state in detail. Next, we introduce the formation of Controlled-Z gate and Hadamard gate in a 4-qubit system using multi-coin quantum walk and then generate 4-qubit cluster state with the composed quantum circuit. Finally, we make a summary of our work.

## 2. Generating Cluster States through Multi-Coin Quantum Walk

Before introducing the generation process, we briefly describe the process of multi-coin quantum walk in an arbitrary graph with $m(m \geq 2)$ nodes. The whole evolutionary Hilbert space is $\mathcal{H} = \mathcal{H}^P \otimes \mathcal{H}^{C_1} \cdots \otimes \mathcal{H}^{C_k}$, where $\mathcal{H}^P$ and $\mathcal{H}^{C_j}$ represent the position space and the $j$-th coin space, respectively. At $j$-th step of quantum walks, the corresponding unitary operator is described as

$$U_j = S_j \cdot (C_j \otimes I), \tag{1}$$

where $j$ implies the $j$-th step of quantum walk while $S$ and $C$ are conditional shift operator and coin operator, respectively. Here, $C_j$ acts on the $j$-th coin space $\mathcal{H}^{C_j}$ and $S_j$ acts on the combination space of position $\mathcal{H}^P$ and $j$-th coin space $\mathcal{H}^{C_j}$. Assume that the initial state is $|\psi(0)\rangle$, then the system state after $k$ steps becomes

$$|\psi(k)\rangle = (U_k U_{k-1} \cdots U_1)|\psi(0)\rangle = (\prod_{j=1}^{k} U_j)|\psi(0)\rangle. \tag{2}$$

where $U_j = (S_j \otimes I^{\mathcal{H}^{C_1}} \cdots \otimes I^{\mathcal{H}^{C_{j-1}}} \otimes I^{\mathcal{H}^{C_{j+1}}} \cdots I^{\mathcal{H}^{C_k}}) \cdot (I^{\mathcal{H}^P} \otimes I^{\mathcal{H}^{C_1}} \cdots \otimes I^{\mathcal{H}^{C_{j-1}}} \otimes C_j \otimes I^{\mathcal{H}^{C_{j+1}}} \cdots I^{\mathcal{H}^{C_k}})$ [14]. Conditional shift operator $S^{n-com}$ in an n-complete graph [16] with $m$ nodes can be described as

$$S^{n-com} = \sum_{x,i=0}^{n-1} |(x+i) \bmod n\rangle\langle x| \otimes |i\rangle\langle i|, \tag{3}$$

and when $m = 2$, corresponding $S^{2-com}$ can be simplified to

$$S^{2-com} = (|0\rangle\langle 0| \otimes |1\rangle\langle 1|) \otimes |0\rangle\langle 0| + (|1\rangle\langle 0| \otimes |0\rangle\langle 1|) \otimes |1\rangle\langle 1|. \tag{4}$$

### 2.1. GHZ States Generation

Before we generate 4-qubit cluster state, it is necessary to generate 3-qubit GHZ state through quantum walk first. Therefore, we consider the generation of GHZ state using 2 pairs of Bell states. The system state of 4 qubits is

$$|\varphi(0)\rangle = \frac{1}{2}(|00\rangle + |11\rangle)_{01}(|00\rangle + |11\rangle)_{23} \tag{5}$$

and our target is to generate GHZ state between qubit 0, 2, and 3, which means that our method is deterministic rather than post-selection based. In the case of 2-complete graph, we only need to go through two quantum walks to obtain the GHZ state, and its evolution is shown in Figure 1, where the circle represents the qubit and the straight line represents the entanglement. Through quantum walks described in Equations (6) and (7), GHZ state formed between qubit 0, 2, and 3 can be obtained. In the implementation of 2-step quantum walks, qubit 0 serves as a walker and qubit 1, 2, and 3 as coins.

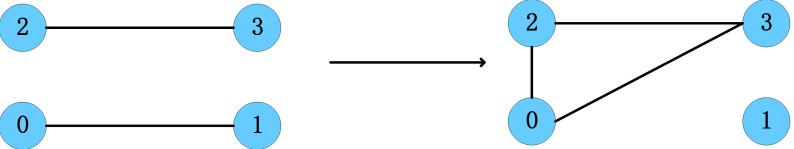

**Figure 1.** Process of generating GHZ state with 2 pairs of Bell states through quantum walk.

$$|\varphi(1)\rangle = S_1^{2-com}(C_1 \otimes I)|\varphi(0)\rangle = \frac{1}{2}(|00\rangle + |01\rangle)_{01}(|00\rangle + |11\rangle)_{23} \tag{6}$$

$$|\varphi(2)\rangle = S_2^{2-com}(C_2 \otimes I)|\varphi(1)\rangle = \frac{1}{2}(|000\rangle + |111\rangle)_{023}(|0\rangle + |1\rangle)_1 \tag{7}$$

In the first step (Equation (6)), to be specific, we take qubit 0 as walker and qubit 1 as coin whose operator is $C_1 = I$. In the second step (Equation (7)), we also take qubit 0 as walker but qubit 2 as coin whose operator is $C_2 = I$. After performing X-basis measurement on qubit 1, we can achieve GHZ state between qubit 0, 2, and 3. This process is simulated using IBM Quantum Experience and the result is shown in Figure 2. It can be seen that GHZ states are generated between targeted qubits. This also proved that GHZ states can be prepared through quantum walk only by using Bell states.

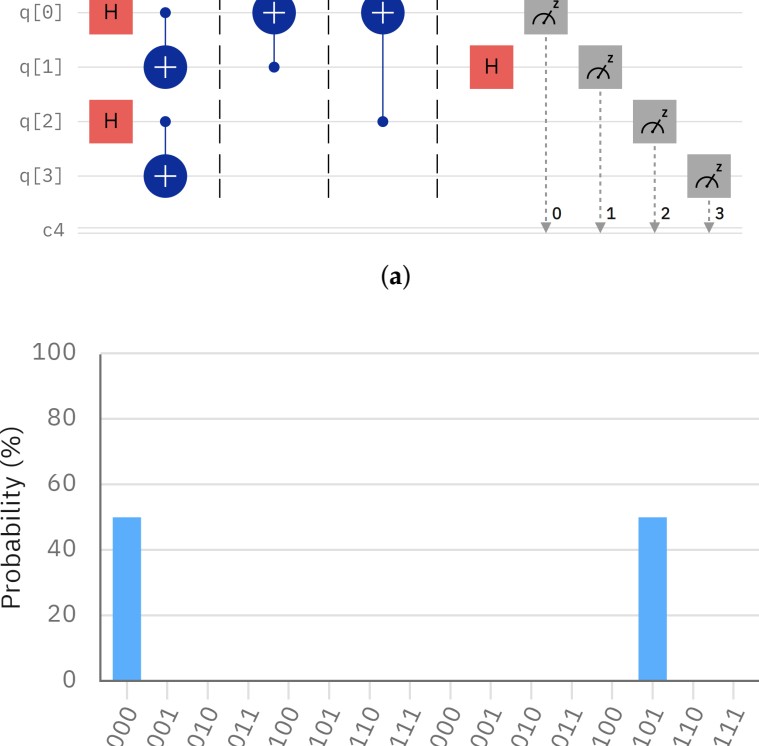

**Figure 2.** Generating GHZ states through 2-step multi-coin quantum walk: (**a**) Quantum circuit generating GHZ states and (**b**) simulation results. CNOT gates are indicated by blue circles in the circuit.

### 2.2. 4-Qubit Cluster States Generation

The 4-qubit cluster states can be described in the form below [17]:

$$|\psi\rangle_{0123} = \frac{1}{2}(|0000\rangle + |0011\rangle + |1100\rangle - |1111\rangle)_{0123} \tag{8}$$

Based on the GHZ state achieved above, we can introduce another pair of Bell state to obtain 4-qubit cluster state through quantum walk, the process of which is shown in Figure 3, where the circle represents the qubit, and the straight line represents the entanglement. Again, the result is determined and we want to form cluster state between qubit 0, 2, 3, and 5 through multi-coin quantum walk in a 2-complete graph.

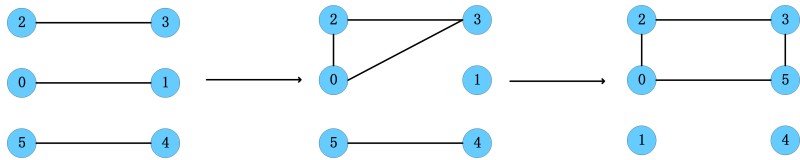

**Figure 3.** Process of generating 4-qubit cluster state with Bell states through quantum walk.

The initial state of the system is composed of 3 pairs of Bell states and can be described as:

$$|\psi(0)\rangle = \frac{1}{2\sqrt{2}}(|00\rangle + |11\rangle)_{01}(|00\rangle + |11\rangle)_{23}(|00\rangle + |11\rangle)_{45}. \tag{9}$$

We first repeat the 2-step quantum walk mentioned above to generate GHZ state (11) and (12), during which qubit 0 serves as a walker and qubit 1 and 2 as coins in turn with $C_1 = I$ and $C_2 = I$. Then, the newly introduced Bell state steps in. In the third step, we take qubit 5 as walker and qubit 4 as coin described as $C_3 = I$. In the fourth step, we also take qubit 5 as walker but qubit 0 as coin with $C_4 = H_0$, where $H_0$ is Hadamard gate performed on qubit 0:

$$H_0 = \frac{1}{\sqrt{2}}\begin{bmatrix} 1 & 1 \\ 1 & -1 \end{bmatrix} \tag{10}$$

For the system state, each step of quantum walk can be described as:

$$|\psi(1)\rangle = \frac{1}{2\sqrt{2}}(|00\rangle + |01\rangle)_{01}(|00\rangle + |11\rangle)_{23}(|00\rangle + |11\rangle)_{45} \tag{11}$$

$$|\psi(2)\rangle = \frac{1}{2\sqrt{2}}(|0\rangle + |1\rangle)_1(|000\rangle + |111\rangle)_{023}(|00\rangle + |11\rangle)_{45} \tag{12}$$

$$|\psi(3)\rangle = \frac{1}{2\sqrt{2}}(|0\rangle + |1\rangle)_1(|000\rangle + |111\rangle)_{023}(|00\rangle + |01\rangle)_{45} \tag{13}$$

$$|\psi(4)\rangle = \frac{1}{4}(|0000\rangle + |0011\rangle + |1100\rangle - |1111\rangle)_{0235}|+\rangle_1|+\rangle_4 \tag{14}$$

After four steps of quantum walk, we perform X-basis measurement on qubit 1 and 4 and a 4-qubit cluster state between qubit 0, 2, 3, and 5 are achieved using only Bell states.

Same as GHZ states, we also use IBM Quantum Experience to verify our method. The corresponding operations are implemented in quantum circuit and shown in Figure 4. We can see that a 4-qubit cluster state are obtained between q[0], q[2], q[3], and q[5].

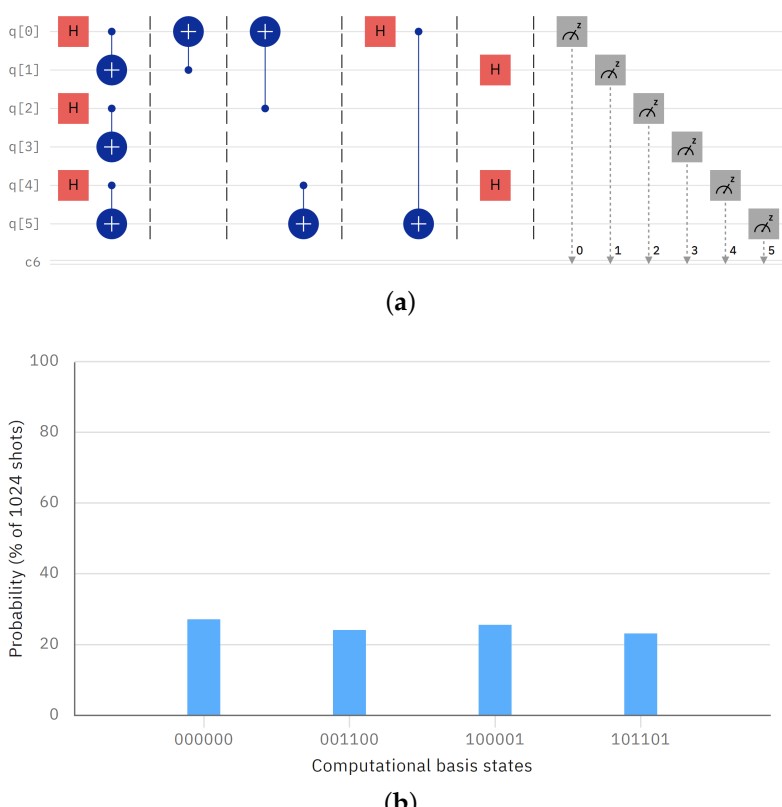

**Figure 4.** Generating cluster states through 4-step multi-coin quantum walk: (**a**) Quantum circuit generating cluster states and (**b**) simulation results.

In this part, 4-qubit cluster states are generated by multi-coin quantum walk in a 2-complete graph using only Bell states and local measurements. This result can also prove that we can generate higher-dimensional entangled qubits following the the same method by adding more pairs of Bell state. Take 5-qubit cluster state as an example: In order to achieve the entangled state $|\psi\rangle = \frac{1}{2}(|00000\rangle + |00111\rangle + |11101\rangle + |11010\rangle)$, a possible 8-step quantum walk can be implemented using 4 pairs of Bell states. The initial state of the system is

$$|\psi(0)\rangle = \frac{1}{4}(|00\rangle + |11\rangle)_{01}(|00\rangle + |11\rangle)_{23}(|00\rangle + |11\rangle)_{45}(|00\rangle + |11\rangle)_{67} \qquad (15)$$

Firstly, we take qubit 0 as walker and qubit 2 and 3 as coins in turn with $C_1 = I$ and $C_2 = I$. Then, Hadamard gate $C_3 = H$ is performed on qubit 0. Next, we take qubit 5 as walker and qubit 3, 4 and 2 as coins, in turn, with $C_4 = I$, $C_5 = I$, and $C_7 = I$. Finally, we take qubit 6 as walker and qubit 7 and 3 as coins with $C_6 = I$ and $C_8 = I$. The process will become more complicated as the dimension goes higher but the advantage of using only Bell states and local measurements remain the same.

This method can also be used in some applications to simplify the process, for example quantum secure direct communication (QSDC) [18]. A brief procedure of QSDC based on our method can be described as follows:

$$|\phi^+\rangle = \frac{1}{\sqrt{2}}(|00\rangle + |11\rangle) \to 00$$

$$|\phi^-\rangle = \frac{1}{\sqrt{2}}(|00\rangle - |11\rangle) \to 11$$

$$\quad (16)$$

$$|\psi^+\rangle = \frac{1}{\sqrt{2}}(|01\rangle + |10\rangle) \to 01$$

$$|\psi^+\rangle = \frac{1}{\sqrt{2}}(|01\rangle - |10\rangle) \to 10$$

Provided that the encoding scheme in Equation (16) is used by the sender Alice and the receiver Bob:

Firstly, Alice prepared an ordered sequence of Bell states of $|\phi^+\rangle$, where every 3 Bell states are seen as a group. For each group, qubit 0, 2, and 4 are kept by Alice while qubit 1, 3, and 5 are sent to Bob, creating sequence $P_{024}$ and $P_{135}$, respectively. After Bob receives sequence $P_{135}$, Alice and Bob carry out 4-step quantum walk using our method to generate cluster states between qubit 0, 2, 3, and 5. X-basis measurement are made on qubit 1 and 4, which serves as the first security check. If both measurement results turn out to be $|0\rangle$, the channel is secure and the corresponding sequences $P'_{02}$ and $P'_{35}$ (the ordered sequences after measurements) are kept, otherwise they are discarded.

Secondly, sufficient groups of qubits are chosen from $P'_{02}$ as checking qubits and one of the operations in $\{I, \sigma_x, \sigma_y, \sigma_z\}$ is performed on each of the checking qubit. The information to be sent is then encoded onto the remaining groups in $P'_{02}$ and sent to Bob. Extra disturbing qubits are added during this process and the positions of disturbing qubits and checking qubits are also told to Bob, as well as the operations carried out on checking qubits and the corresponding results.

Next, Bob removes the disturbing qubits after receiving the sequence $P'_{02}$ and then measure the checking qubits and compare the result to what Alice has told Bob, which is the second security check. If the results are the same, the channel is determined to be secure and the checking qubits can also be removed. Otherwise, the channel is discarded and another one needs to be established.

Finally, Bob can restore the information from cluster states in $P''_{02}$ and $P''_{35}$ (the sequences after second security check) according to the encoding scheme.

Compared with original QSDC, cluster states can now be generated during the process rather than beforehand and QSDC can be performed without adding extra qubits as checking qubit or disturbing qubit for the first security check because there are qubits left (qubit 1 and 4) after generating the cluster states. In addition, only X-basis measurements are required in the first security check which slightly simplifies the process.

## 3. Generating Cluster States Using Quantum Gates Formed by Multi-Coin Quantum Walk

Multi-coin quantum walk can also construct quantum gates which are the basis of quantum circuit. Therefore, we take one step further to generate 4-qubit cluster states using quantum gates formed by quantum walk in order to introduce a more practical method of cluster state generation. It is the most common case to use Controlled-Z gates (CZ gates) to generate 4-qubit cluster states: 4 qubits with an initial state of $|+\rangle_1|+\rangle_2|+\rangle_3|+\rangle_4$ can be converted to 4-qubit cluster state through 3 CZ gates and 2 Hadamard gates, which is shown in Figure 5 and system states after each operation are described as below:

$$|\Phi(1)\rangle = |+\rangle_1 |+\rangle_2 |+\rangle_3 |+\rangle_4 \tag{17}$$

$$|\Phi(2)\rangle = U_{CZ}^{12}|\Phi(1)\rangle = \frac{1}{\sqrt{2}}(|0\rangle|+\rangle + |1\rangle|-\rangle)_{12}|+\rangle_3|+\rangle_4 \tag{18}$$

$$|\Phi(3)\rangle = U_H^2|\Phi(2)\rangle = \frac{1}{\sqrt{2}}(|0\rangle|0\rangle + |1\rangle|1\rangle)_{12}|+\rangle_3|+\rangle_4 \tag{19}$$

$$|\Phi(4)\rangle = U_{CZ}^{23}|\Phi(3)\rangle \tag{20}$$

$$= \frac{1}{2}(|0\rangle|0\rangle|0\rangle + |0\rangle|0\rangle|1\rangle + |1\rangle|1\rangle|0\rangle - |1\rangle|1\rangle|1\rangle)_{123}|+\rangle_4$$

$$|\Phi(5)\rangle = U_H^4 U_{CZ}^{34}|\Phi(4)\rangle \tag{21}$$

$$= \frac{1}{2}(|0\rangle|0\rangle|0\rangle|0\rangle + |0\rangle|0\rangle|1\rangle|1\rangle + |1\rangle|1\rangle|0\rangle|0\rangle - |1\rangle|1\rangle|1\rangle|1\rangle)_{1234}$$

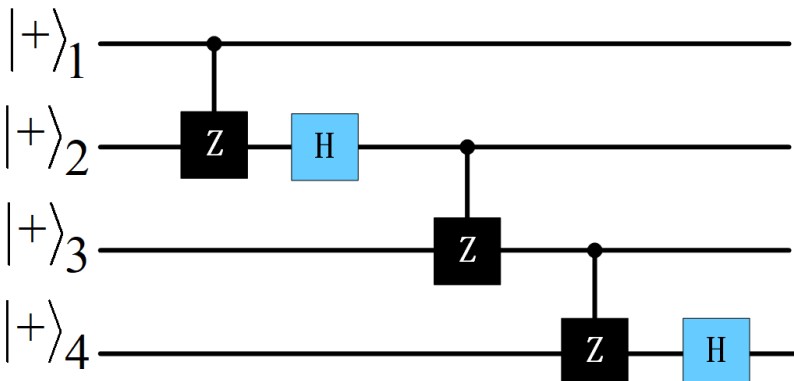

**Figure 5.** Process of generating 4-qubit cluster state using several quantum gates.

Then, we introduce how to implement CZ gate and Hadamard gate using quantum walk. Previous studies [15] describe the 2-qubit and 3-qubit systems in detail, but only give a basic idea of the 4-qubit system. Here, we propose a complete and improved method of quantum gate implementation in 4-qubit system. The idea is to indicate qubits using computational basis and to simulate general computing gates using controlled conversion operators. For 4-qubit quantum walk of a single particle, the qubit 1 is considered as walker while the other 3 qubits describe the position in Hilbert space. Therefore, the process of 4-qubit quantum walk of a single particle can be mapped onto a 3D cube (shown in Figure 6) and qubit 1 (real particle) will perform quantum walk within 8 positions according to the operation to be implemented.

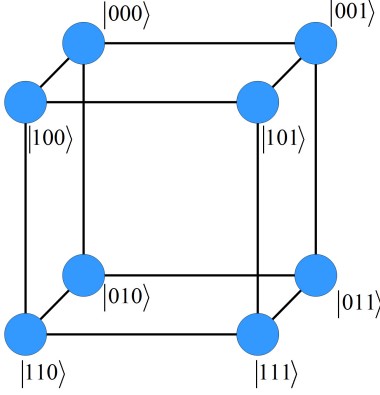

**Figure 6.** Mapping 4-qubit quantum walk onto a 3D cube.

As for the operators, the coin operator of the 4-qubit quantum walk is identical to the one of the 2-qubit or 3-qubit quantum walks [15], which are generally described as:

$$C(\epsilon, \alpha, \theta) = \begin{bmatrix} e^{i\epsilon}cos(\theta) & e^{i\alpha}sin(\theta) \\ e^{-i\alpha}sin(\theta) & -e^{-i\epsilon}cos(\theta) \end{bmatrix}, \tag{22}$$

while shift operator $S$ is more complicated. In order to cope with the extra qubit (extra dimension in position space), shift operator $S$ is different according to different transition traces. If qubit 2 (first part of position) is fixed, in which case the state transition takes place in the front and back side of the cube in Figure 6, $S = I$; if qubit 4 (third part of position) is fixed, in which case the state transition takes place in the left and right side of the cube in Figure 6, $S = -I$. Therefore, we have

$$\begin{cases} S = I, & when \quad |000\rangle \to |001\rangle \to |011\rangle \to |010\rangle \\ & and \quad |100\rangle \to |101\rangle \to |111\rangle \to |110\rangle \quad (trace1), \\ S = -I, & when \quad |000\rangle \to |010\rangle \to |110\rangle \to |100\rangle \\ & and \quad |001\rangle \to |011\rangle \to |111\rangle \to |101\rangle \quad (trace2) \end{cases} \tag{23}$$

Based on this, we can build the required CZ gate and Hadamard gate.

*3.1. CZ Gate*

CZ gate can be described as:

$$U_{CZ} = \begin{bmatrix} 1 & 0 & 0 & 0 \\ 0 & 1 & 0 & 0 \\ 0 & 0 & 1 & 0 \\ 0 & 0 & 0 & -1 \end{bmatrix} \tag{24}$$

According to the mapping mentioned above, the qubit 2, 3, and 4 of a 4-qubit system are mapped to the position space and a position-dependent coin operator is implemented on the walker. Position-dependent coin means that for the same operation (such as CZ gate), the actual coin operator performed on the walker (qubit 1) is different according to which position the walker occupies. The possible operators required to form a CZ gate can be described as:

$$\begin{cases} Z = \sigma_z \otimes I_p, \\ I = I \otimes I_p, \\ V = e^{i\pi}I \otimes I_p, \end{cases} \tag{25}$$

where $I_p$ is the identity operator in the position space. We can then map the CZ gate onto 8 positions in the position space, as shown in Figure 7. The directions of the arrows are determined by the shift operators, which only indicates how the walker moves instead of the state conversion.

Take Figure 7a as an example. It describes the CZ gate between qubit 1 and 2, namely $U_{CZ}^{12}$. If qubit 1 (the walker) is at $|000\rangle$, then 2 possible operations might be carried out: if the walker follows trace 1, then coin operator $C = I$ and shift operator $S = I$ are performed on the walker, implementing a CZ gate and making the walker move from $|000\rangle$ to $|001\rangle$; if the walker follows trace 2, then coin operator $C = -I$ and shift operator $S = -I$ are performed on the walker, also implementing a CZ gate and making the walker move from $|000\rangle$ to $|010\rangle$.

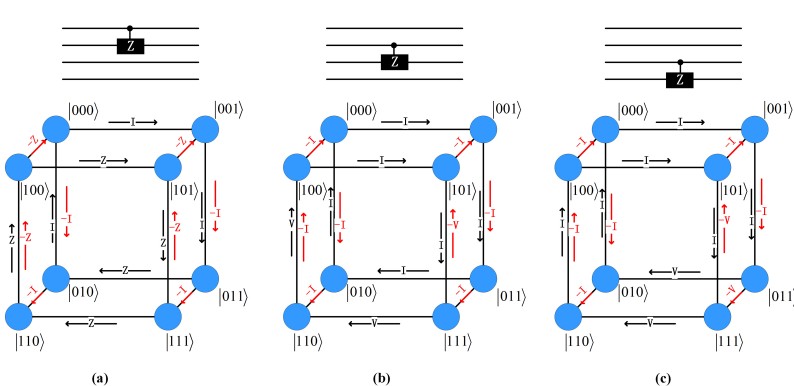

**Figure 7.** Implementation of (**a**) $U_{CZ}^{12}$, (**b**) $U_{CZ}^{23}$ and (**c**) $U_{CZ}^{34}$ through 4-qubit quantum walk. Qubit 2, 3 and 4 are used to determine the 8 positions in Hilbert space and actual CZ operation is performed on qubit 1. The operator on the arrow indicates the operation performed on the walker if the walker occupied the starting position of the arrow. The black arrows correspond to trace 1 while the red arrows correspond to trace 2.

### 3.2. Hadamard Gate

In order to build Hadamard gate, conditional shift operator should be divided into two parts $S_-$ and $S_+$ to move the particle in left and right directions, which can be described as [15]

$$
\begin{aligned}
S_-^i &= \sum_{m \in Z, i} (|i\rangle\langle i| \otimes |m-1\rangle\langle m| + |j \neq i\rangle\langle j \neq i| \otimes |m\rangle\langle m|), \\
S_+^j &= \sum_{m \in Z, j} (|j\rangle\langle j| \otimes |m+1\rangle\langle m| + |i \neq j\rangle\langle i \neq j| \otimes |m\rangle\langle m|),
\end{aligned}
\tag{26}
$$

where $|i\rangle, |j\rangle \in \{|0\rangle, |1\rangle\}$ are the basis of Hilbert space and $m \in \mathbb{Z}$ indicates the possible position in Hilbert space (in our case $m \in [0,7]$). The overall transition operator of Hadamard gate is

$$
H_1 = I(C(0,0,\pi/4) \otimes I_p)
\tag{27}
$$

where $C(0,0,\pi/4)$ is the coin operator given in Equation (22). The Hadamard gate is implemented by combining $S_-$ and $S_+$ differently to evolve the coin state of the particle in superposition of position space which is composed of qubit 2, 3, and 4. It is a position-dependent coin operation followed by shift operators.

The implementation of Hadamard gate also depends on the qubit it is performed on. For $U_H^2$ (as shown in Figure 8a), the 8 positions in Hilbert space are marked as:

$$
\begin{aligned}
|000\rangle &\to l = 0, & |100\rangle &\to l = 4 \\
|001\rangle &\to l = 1, & |101\rangle &\to l = 5 \\
|011\rangle &\to l = 2, & |111\rangle &\to l = 6 \\
|010\rangle &\to l = 3, & |110\rangle &\to l = 7
\end{aligned}
\tag{28}
$$

The quantum walk operations to be carried out following trace 1 ($l = 0 \to 1 \to 2 \to 3$ or $l = 4 \to 5 \to 6 \to 7$, shown as black arrows in Figure 8a) are:

$$
\begin{aligned}
H_+^0 |i\rangle \otimes |m\rangle &= \sigma_x^m S_+^i (\sigma_x \otimes I) \\
H_+^1 |i\rangle \otimes |m\rangle &= \sigma_x^m S_+^i (\sigma_z \otimes I) \\
H_-^0 |j\rangle \otimes |m\rangle &= \sigma_x^m S_-^j (\sigma_x \otimes I) \\
H_-^1 |j\rangle \otimes |m\rangle &= \sigma_x^m S_-^j (\sigma_z \otimes I)
\end{aligned}
\tag{29}
$$

where $\sigma_x^m = \sigma_x \otimes |m\rangle\langle m| + I \otimes \sum |p\rangle\langle p|$ and $|m\rangle$ is the initial position. $p = l + 4$ when the walker is on $|000\rangle$ to $|010\rangle$ circle and $p = l - 4$ when the walker is on $|100\rangle$ to $|110\rangle$ circle ($l \neq m$ for both conditions).

The quantum walk operations to be carried out following trace 2 ($l = 0 \rightarrow 3 \rightarrow 7 \rightarrow 4$ or $l = 1 \rightarrow 2 \rightarrow 6 \rightarrow 5$, shown as red arrows in Figure 8a) are:

$$
\begin{aligned}
\hat{H}_+^0 |i\rangle \otimes |m\rangle &= \sigma_x^m (-S_+^i)(-\sigma_x \otimes I) \\
\hat{H}_+^1 |i\rangle \otimes |m\rangle &= \sigma_x^m (-S_+^i)(-\sigma_z \otimes I) \\
\hat{H}_-^0 |j\rangle \otimes |m\rangle &= \sigma_x^m (-S_-^j)(-\sigma_x \otimes I) \\
\hat{H}_-^1 |j\rangle \otimes |m\rangle &= \sigma_x^m (-S_-^j)(-\sigma_z \otimes I)
\end{aligned}
\tag{30}
$$

where $\sigma_x^m = \sigma_x \otimes |m\rangle\langle m| + I \otimes \sum_{l \neq m} |l\rangle\langle l|$.

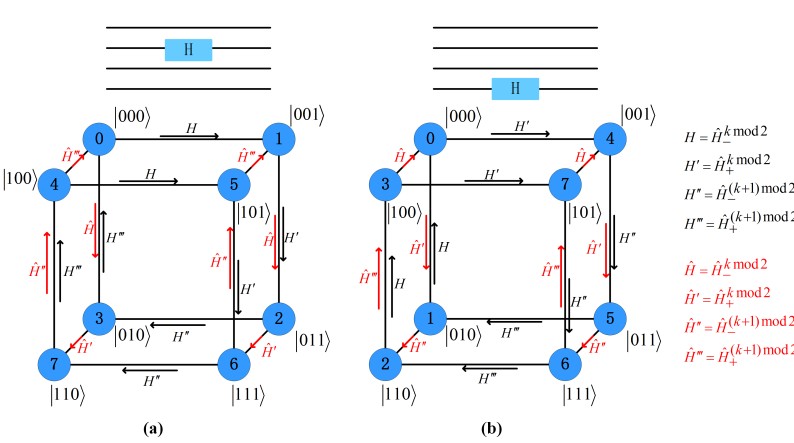

**Figure 8.** Implementation of (**a**) $U_H^2$ and (**b**) $U_H^4$ through 4-qubit quantum walk. Qubit 2, 3, and 4 are used to determine the 8 positions in Hilbert space and actual Hadamard operation is performed on qubit 1. The operator on the arrow indicates the operation performed on the walker if the walker occupied the starting position of the arrow. Additionally, the specific operations are listed in detail on the right side. The black arrows correspond to the trace 1 while the red arrows correspond to the trace 2.

For $U_H^4$ (as shown in Figure 8b), the 8 positions in Hilbert space are marked as:

$$
\begin{aligned}
|000\rangle \rightarrow l = 0, \quad & |001\rangle \rightarrow l = 4 \\
|010\rangle \rightarrow l = 1, \quad & |011\rangle \rightarrow l = 5 \\
|110\rangle \rightarrow l = 2, \quad & |111\rangle \rightarrow l = 6 \\
|100\rangle \rightarrow l = 3, \quad & |101\rangle \rightarrow l = 7
\end{aligned}
\tag{31}
$$

The quantum walk operations to be carried out following trace 1 ($l = 0 \rightarrow 4 \rightarrow 5 \rightarrow 1$ or $l = 3 \rightarrow 7 \rightarrow 6 \rightarrow 2$, shown as black arrows in Figure 8b) are the same as Equation (30). Although the quantum walk to be carried out following trace 2 ($l = 0 \rightarrow 1 \rightarrow 2 \rightarrow 3$ or $l = 4 \rightarrow 5 \rightarrow 6 \rightarrow 7$, shown as red arrows in Figure 8b) are the same as Equation (29).

Now we have successfully mapped the Hadamard gate to 8 positions in Hilbert space. Same as above, the directions of the arrows result from the shift operator and only indicate the movement of the walker in position space rather than state conversion. The actual operation on the cube of Hadamard gate is similar to the one of CZ gate. Position-dependent coin operator $C$ and shift operator $S$ will be determined by not only the position of the walker and the trace it chooses but also the targeted qubit. After the operation, Hadamard gate is performed on qubit 1 (walker), and it will move to next position according to the shift operator.

### 3.3. Generating Cluster States

So far we have implemented the required CZ gate and Hadamard gate by mapping the corresponding quantum walk onto a 3D cube. Now we can combine the CZ gates and the Hadamard gates and carry out the 4-step operations mentioned in Equations (17)–(21). Figure 9 shows the equivalent quantum walk scheme for 4-qubit quantum circuit. The system state remains the same from $|\Phi(1)\rangle$ to $|\Phi(2)\rangle$ and from $|\Phi(3)\rangle$ to $|\Phi(4)\rangle$, so the corresponding cube do not change during these two steps. We can see that 4-qubit cluster state is achieved after 5 coin operations: the qubit at $|000\rangle$, $|100\rangle$, $|011\rangle$ and $|111\rangle$ are $|0\rangle$, $|1\rangle$, $|0\rangle$ and $|1\rangle$, respectively, which gives $|0\rangle_1|000\rangle_{234}$, $|1\rangle_1|100\rangle_{234}$, $|0\rangle_1|011\rangle_{234}$ and $|1\rangle_1|111\rangle_{234}$.

Our improved method of simulating quantum gates using multi-coin quantum walk can potentially be used to generate higher entangled qubits. However, the mapping scheme can be different depending on the targeted qubits. Some possible schemes have been shown in [15].

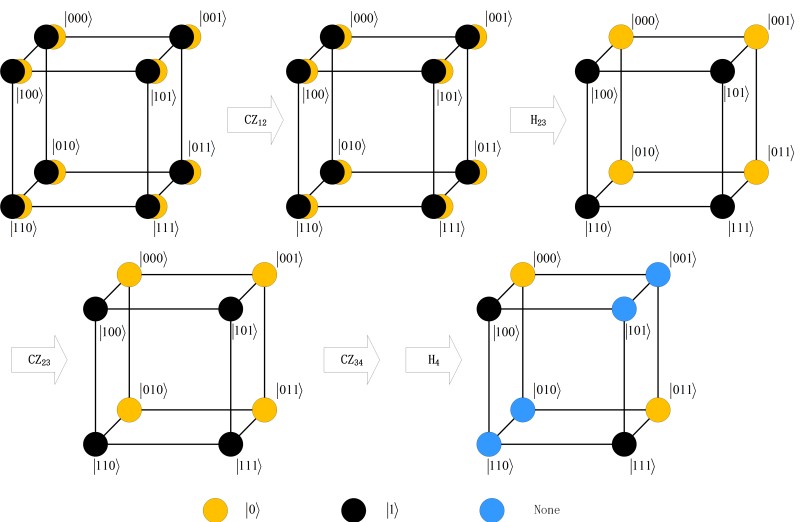

**Figure 9.** Generating 4-qubit cluster state using single particle quantum walk, where yellow circles represent $|0\rangle$, black circles represent $|1\rangle$, and blue circles represent vacancies.

### 4. Conclusions

In this paper, we propose two methods of generating 4-qubit cluster state between targeted qubits. We first extend former researches to a 4-qubit system and use multi-coin quantum walk to generate 4-qubit cluster state with only 3 pairs of Bell states and local measurement. Compared with traditional optical methods, this method simplifies the generation process and lower the requirement of resources. Furthermore, we present another method of generating 4-qubit cluster states by building quantum gates using multi-coin quantum walk. We improve the quantum gate implementation described in former works and make it suitable for the 4-qubit system, based on which we form the targeted quantum circuit. This can be helpful in real quantum computing scenario. For future researches, we think these methods can be extended to generate entangled states with more qubits. For the first method, we can simply add extra Bell states to the current entangled state and carry out additional quantum walk. New coin qubit and walker qubit are chosen based on the targeted entangled states, following the same method for 3-qubit and 4-qubit entangled states. It is more complicated for the second method since a different mapping scheme might be required for more qubits. In addition, we do not take into consideration the noise of the channel or any decoherence and depolarization of the channel. Studies on the consumption rate of entanglement, transmission rate, and complexity of the execution can also be done in future works.

**Author Contributions:** Conceptualization, T.W. ; methodology, T.W.; software, T.W.; validation, T.W.; formal analysis, T.W.; investigation, T.W.; resources, T.W. and J.L.; data curation, T.W.; writing—original draft preparation, T.W.; writing—review and editing, T.W. and J.L.; visualization, T.W.; supervision, X.C.; project administration, X.C.; funding acquisition, X.C. All authors have read and agreed to the published version of the manuscript.

**Funding:** This research received no external funding.

**Institutional Review Board Statement:** Not applicable.

**Informed Consent Statement:** Not applicable.

**Data Availability Statement:** Not applicable.

**Conflicts of Interest:** The authors declare no conflict of interest.

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
