# Peer review of "Four-Qubit Cluster States Generation through Multi-Coin Quantum Walk"

_applsci, doi:10.3390/app12178750_

Round 1

Author Response

Thank you for your precious comments and suggestions. The revised paper has been uploaded. Here are the major changes:

Most of the introduction has been rewritten since the original one is not clear and slightly off the point. Former study on multi-coin quantum walk are emphasized and our contribution are presented in a more explicit way both in the introduction (line 78-88) and in the main context (line 168-170, 173-175). The qubits in section 2 are now labelled from 0 (both in the equations and the figures) since the original labels can be confusing. In the conclusion part, a clearer summary of our work and discussion on a possible general method of generating larger number of qubits are added. Some changes are made to the references with one reference added and two reference deleted and the numbers are changed due to rewritten context in Introduction part.

(Sorry that strikethrough in subscript and some equations are not displayed properly.)

As for your suggestion that we should present the different between our work and former researches, clearer statements are added as mentioned above. We hope this can  address the problem.

Again, thank you for your comments and suggestions!

Reviewer 2 Report

Unfortunately, the manuscript shows rather weak acquaintance of the authors with the theme. It is manifested in a rather standard way. At the beginning the authors address difficulties without understanding their nature, then put in a number of supposedly complicated and respectfully looking formulas, and than present something rather trivial simultaneously trying to wrap it in nice-looking formulas and pictures. 

Essence of this text is just a few simple unitary transformations of few qubit system. There is nothing and interesting here. There is no new physics and insight. There is nothing to simulate using "IBM Quantum Experience". A scrap of paper would be quite sufficient.

This text cannot be published. 

Author Response

Thank you for your precious comments and suggestions. The revised paper has been uploaded. Here are the major changes:

Most of the introduction has been rewritten. Former study on multi-coin quantum walk are emphasized and our contribution are presented in a more explicit way both in the introduction (line 78-88) and in the main context (line 168-170, 173-175). The qubits in section 2 are now labelled from 0 (both in the equations and the figures) since the original labels can be confusing. In the conclusion part, a clearer summary of our work and discussion on a possible general method of generating larger number of qubits are added. Some changes are made to the references with one reference added and two reference deleted and the numbers are changed due to rewritten context in Introduction part.

(Sorry that strikethrough in subscript and some equations are not displayed properly.)

As for your comment on our work. Firstly, we must admit that we didn’t present it in a clear way, especially in the Introduction part. The original introduction is slightly off the point, so we rewrite most of it. Secondly, we believe that our work is meaningful because we managed to propose two feasible and efficient methods to generate 4-qubit cluster states compared to traditional methods. We extended the method proposed in Ref.14 (Ref.9 in the original paper) and generate larger number of qubits. We also improved the implementation of quantum gate in Ref.15 (Ref.18 in the original paper) since Ref.15 didn’t present a practical implementation for 4-qubit system. We have added these explanations to the revised paper. Besides, applying quantum walk to qubit manipulation and quantum computing is widely accepted and discussed. The calculation may not be complicated (as far as in our paper), but the idea of using quantum walk to finish specific analysis and operations is worth considering.

Again, thank you for your comments and suggestions! 

Reviewer 3 Report

Dear Editor,

In the present manuscript, the authors proposed schemes to generate four qubit cluster states via multi-coin quantum walks. While the original idea of this work is based on Ref. [9], conserving the importance of multi qubit cluster states in quantum information processing, I think it presents important case study. It also presents quantum gate implementation via quantum walks which could be useful for other quantum information processing. With some revisions below, I would like to recommend this manuscript for publication.

  1. While this manuscript only reports four qubit cluster states generation, it is of importance to know general aspect of the scheme. Can it be generalized to larger number of qubits? Please discuss possibility to the possible expansion. 

  2. The scheme is not post-selection based, but deterministic, right? Readers can easily misunderstand the scheme is post-selection based since there are qubits which does not form entanglement with others. Please specifically mention this.

  3. I suggest some relevant references about entanglement generation and measurement  via quantum walk, e.g., Phy. Lett. A 381, 3875-3879 (2017), Opt. Express 26, 29539 (2018), Opt. Express 30, 30525 (2022). While the last two papers present entanglement via spatial overlap, it also can be interpreted as quantum walks with coin and walker states as polarization and spatial modes, respectively.

  4. While the qubit labels in the conceptual diagram begins with 1, the qubit labels in quantum circuits begin with q[0]. It is very confusing to follow the manuscript, and the notation should be unified.

  5. There are some representation errors.
    - Some equations are not normalized, e.g., Eq. (9) and (10).
    - The descriptions in line 100~102 seems to have error. As I understand, for the first step, coin should be qubit 2 not 3. And for the second step, coin should be qubit 3 not 2.
    - In Eq. (17), the cluster state is formed among 1346 not 1345.

Author Response

Thank you for your precious comments and suggestions. The revised paper has been uploaded. Here are the major changes:

Most of the introduction has been rewritten since the original one is not clear and slightly off the point. Former study on multi-coin quantum walk are emphasized and our contribution are presented in a more explicit way both in the introduction (line 78-88) and in the main context (line 167-171, 173-175). The qubits in section 2 are now labelled from 0 (both in the equations and the figures) since the original labels can be confusing. In the conclusion part, a clearer summary of our work and discussion on a possible general method of generating larger number of qubits are added. Some changes are made to the references with one reference added and two reference deleted and the numbers are changed due to rewritten context in Introduction part.

(Sorry that strikethrough in subscript and some equations are not displayed properly.)

As for the specific points you have mentioned:

  1. Discussions on the possible generalization of our methods are added. We are sure that the methods can be applied to larger number of qubits. The first method can follow exactly the same way by adding new pairs of Bell states, while the second method might need another mapping scheme according to the targeted entangled qubits.
  2. Yes, the scheme is deterministic. Explanations on the result has been added for GHZ state (line 126-127) and cluster state (line 148-150).
  3. Thank you so much for your suggestion. We found these references very useful, especially Phy. Lett. A 381, 3875-3879 (2017) since we use a similar method of carrying out coined quantum walk. The other two papers might be helpful for our future research but more time is needed.
  4. The labels are counted from 0 now. We hope this can make it less confusing.
  5. These representation errors have been corrected.

Again, thank you for your comments and suggestions!

Round 2

Reviewer 2 Report

I am thankful to the authors for their attempt to improve the paper. Now the manuscript looks a bit more honest. However, this reveals to even larger degree triviality of results. The authors do not actually suggest a method as something practical. They just show some simple unitary transformations needed to create a state in question. By the way, they did not answer what and why exactly was necessary to verify using  IBM Quantum Experience. 

I still do not consider a result as something significant to publish in MDPI "Applied Sciences". I also question adequacy of the manuscript for "Applied Sciences". "Quantum Reports" is a better venue for this paper. 

Author Response

Thank you for your comments and suggestions. The newly revised version has been uploaded. We are sorry that we didn’t manage to answer all of your questions. The reason that we use IBM quantum experience is simply that we want to give a clear visual presentation of the generated entangled states, especially for the 4-qubit cluster state case since the qubits can be confusing. We admit that the simulations don’t have a strong relation to our actual method but we think they are helpful for understanding the paper. As for the “practicality”, the original thought of using this term is that our method requires fewer complicated operations so it can be more “feasible” or “easier” than former methods. Now we realized that without certain applications or actual experiments, it’s not very appropriate to call it practical. We changed some of the “practical” expression and tried to add a possible application scenario in Sec.2. We hope this can add some practicality and usability to our method. Besides, we also added some brief introduction of extending the first method to higher-dimensional entangled qubits.

Again, thank you for your comments. They are very helpful for us to polish our paper.